



# Comparison of particle number concentrations measured with AQ Urban sensors in two different environments in Helsinki, Finland

Kimmo Teinilä[1], Teemu Lepistö[2], Jarkko V. Niemi[3], Harri Portin[3], Anssi Julkunen[3], Anu Kousa[3], Joel Kuula[1], Hanna E. Manninen[3], Pasi Aalto[4], Tuukka Petäjä[4], Topi Rönkkö[2], Erkka Saukko[5], and Hilkka Timonen[1]

[1]Atmospheric composition research, Finnish Meteorological Institute, P.O. Box 503, FI-00101, Finland.
[2]Aerosol Physics Laboratory, Physics Unit, Tampere University, P.O. Box 692, FI-33014, Finland.
[3]Helsinki Region Environmental Services Authority HSY, P.O. Box 100, FI-00066, Finland.
[3]Aerosol Physics Laboratory, Physics Unit, Tampere University, P.O. Box 692, FI-33014, Finland.
[4]Institute for Atmospheric and Earth System Research (INAR) / Physics, Faculty of Science, University of Helsinki, P.O. Box 64, FI-00014, Finland.
[5]Pegasor Oy, Hatanpään valtatie, 34, 33100 Tampere, Finland.

*Correspondence to*: Kimmo Teinilä (kimmo.teinila@fmi.fi)

**Abstract.** The use of a diffusion charger based AQ Urban sensors to monitor particle number concentrations was investigated in Helsinki metropolitan area. The comparisons between the AQ Urban sensors and traditional butanol CPCs were made at a heavily trafficked street canyon (Traffic Supersite) and at an urban background site (UB Supersite) in 2022. The agreement with the measured particle number concentrations within different AQ Urban units was good. Comparison of the AQ Urban sensor with the two CPCs showed that AQ Urban sensors should be suitable to measure concentration of particles approx. larger than 10 nm in highly trafficked areas. The long-term agreement between AQ Urban sensors and CPCs was also investigated in the two different environments between January 1st and August 15th, 2022. Overall, the correlation between AQ Urban sensors and the CPCs was good at both sites (*r* being 0.93 and 0.89, respectively). The increased concentration of particles smaller than 10 nm and long-range transported pollution affected the accuracy of AQ Urban sensors. Despite this downside of the method, the correlation between the AQ Urban sensor and the CPCs was good during the whole measurement period, indicating that the sensor is well suitable for long-term particle number concentration monitoring in urban environments in Finland. However, the observed effect of bi-modal particle size distribution suggests that the performance of diffusion charger-based sensors may vary in different geographic regions depending on the regional background concentrations of accumulation mode particles which should be considered when applying the method in different locations.

## 1 Introduction

Exposures to particulate pollutants can cause serious health problems (Atkinson et al., 2014), and exposures to increased levels of particulate pollutants have been estimated to cause 3.3 million premature deaths per year on the global scale (Lelieveld et al., 2015). Fine particles (< 2.5 µm) can be transported deep into the human respiratory tract (Zanobetti et al., 2014) and especially ultrafine particles ($D_p$ < 0.1 µm) can enter even deeper into the respiratory tract (Schraufnagel, 2020). Especially in heavily trafficked environments, like street canyons, the concentration of ultrafine particles can increase significantly causing adverse health effects (Pirjola et al., 2017; Rönkkö and Timonen, 2019; Trechera et al., 2023). For example, Hänninen et al., (2025) suggested that ultrafine particles would be the most significant air pollutant regarding premature deaths in Europe in 2023. In general, however, the health effects of ultrafine particles are not completely understood yet (Vallabani et al., 2023).



The main anthropogenic sources of fine particle pollution in Helsinki metropolitan area are direct vehicular emissions, road dust and residential wood burning (Aurela et al., 2015; Carbone et al., 2014; Järvi et al., 2008; Saarikoski et al., 2008; Savadkoohi et al., 2023). In addition, long-range and regional transport increase the concentration of particulate matter in Helsinki metropolitan area (Niemi et al., 2009, 2005, 2004). Secondary aerosol formation during transportation increases the size of these particles e.g. (Harni et al., 2023). Particle number concentration (PNC) typically increases in heavily trafficked areas in Helsinki metropolitan area e.g. morning and afternoon rush hours. Trapping of pollutants in the boundary layer during cold days also increases PNC. In contrast to regional or long-range transported particles, the increased PNC in heavily trafficked areas is connected to small particle size (Pirjola et al., 2017; Rönkkö et al., 2017). The highest PNC in Helsinki Metropolitan area are typically measured near highways, heavily trafficked streets or at airports (Lepistö et al., 2023).

Due to the harmful health effects of ultrafine particles, WHO has recommended the monitoring of PNC (WHO, 2021). WHO also recommended that the minimum lower limit of particle size should be at least 10 nm for monitoring measurements (WHO, 2021). According to EU directive 2024 PNC was regulated to be monitored at rural and urban background supersites and at "hotspot" sites with high (EU) 2024/2881 (2024). This same directive states that the lower limit of the PNC measurements should be 10 nm, which corresponds to the lower particle size of the CEN standard for outdoor butanol CPC measurements (EN 16976:2024).

Outdoor PNC measurements are typically performed using butanol CPC instruments which are widely used also in laboratories. As PNC has typically high spatial and temporal variation, continuous measurements of PNC by utilizing a wide measurement network could be beneficial especially in big cities. The use of the traditional CPCs has some drawbacks if continuous PNC measurements at many sites are intended to be conducted. The price of the CPC measurements is high (instrument purchase price and service), and the maintenance of the measurements is time consuming (maintenance and frequent butanol addition). Due to the above reasons, the PNC monitoring networks based on CPCs are still quite rare. Measuring devices based on diffusion charging could be useful if the coverage of indicative PNC measurements is wanted to be increased. In earlier studies the PNC measured with diffusion-based instruments has been found to be in the range ±50 % (Todea et al., 2017) and in the range ±30 (Asbach et al., 2024) when comparing to traditional butanol CPCs.

In Helsinki Metropolitan area, diffusion charger-based instruments, (AQ™ Urban sensors, Pegasor Oy, Finland) are used at eight measurement stations to continuously monitor PNC concentration. In addition, these sensors measure the lung-deposited surface area (LDSA) concentration of particles e.g. (Kuula et al., 2020), Since the AQ Urban sensor measurement technique differs from the traditionally used CPCs, we investigate the suitability of AQ Urban for PNC measurements in different urban environments. In this paper we compare the PNC measured with the AQ Urban sensors and CPCs at two sites in Helsinki metropolitan area during 7.5-month measurement period. In addition to this a comparison measurement with seven AQ Urban sensors and two CPCs were made during 6-week measurement period. The study aims to gain better understanding of the potential and challenges of AQ Urban, and diffusion charger-based sensors in general, in long-term PNC monitoring.



## 2   Experimental

The 7.5-month measurement period was conducted at two sites in Helsinki Metropolitan area between January1$^{st}$ and August 15$^{th}$, 2022. The measurement sites were Traffic Supersite and Urban Background Supersite (UB Supersite) in Helsinki. PNC measured with the AQ Urban sensors were compared to those measured with the CPCs during the measurement period at the two sites. In addition, a 6-week comparison measurement with 7 different AQ Urban sensors were made at the Traffic Supersite between August 30$^{th}$ and September 19$^{th}$, 2022.

The Traffic Supersite station is an urban measurement station operated by the Helsinki Region Environmental Services Authority (HSY), located in a street canyon on the street Mäkelänkatu (60.19654° N, 24.95172° E) in Helsinki. The Traffic Supersite station monitors continuously urban air quality with measurements of particulate and gaseous components. The measurement station is markedly affected by motor vehicle emissions since it is a street canyon, consisting of six lanes (Hietikko et al., 2018). More detailed descriptions of the site and its air flow patterns are found in (Barreira et al., 2021; Hietikko et al., 2018; Kuuluvainen et al., 2018).

The Urban Background Supersite (UB Supersite, 60.20306° N, 24.96103° E) is the SMEAR III station located in Kumpula campus area (Järvi et al., 2009). The effect of local traffic is quite low at the UB Supersite compared to the Traffic Supersite because of the markedly longer distance to the main road (approximately 150 m from the station with a daily traffic load of approximately 50 000 vehicles). The UB Supersite is affected by residential wood combustion during the winter months (Järvi et al., 2008). At the UB Supersite particle physical and chemical properties and trace gases are continuously measured.

PNC was measured at the Traffic Supersite and at the UB Supersite with diffusion charger-based AQ Urban (Pegasor, Finland) sensors.  At both sites the lower limit of particle size was adjusted to be 10 nm, while the larger particle detection size of the instrument was ~600 nm. (Kuula et al., 2019; Rostedt et al., 2014). The AQ Urban sensor measures the escaping current of charged particles. The measured escape current of the AQ Urban sensor closely matches the lung deposited surface area of particles and is reported in addition to particle number. The instrument can estimate the count mean diameter (CMD) by continuously stepping between a low and variable, high voltage settings; the median particle size is determined by the cutoff voltage of the half-maximum signal compared to the low cutoff signal. Using this mean particle diameter and assuming a lognormal particle size distribution with fixed standard deviation the instrument calculates the PNC (Janka and Saukko, 2017). The temperature of the AQ Urban sensors was set to be 40 °C above the ambient temperature.

PNC was measured also with CPC instruments at both sites. At the Traffic Supersite the used CPC was an A20 (Airmodus Ltd.) with a cut-size (D$_{p50}$) 5.4 nm and at the UB Supersite the used instrument was a CPC model 3756 (TSI) with a cut-size (D$_{p50}$) 7 nm. A dilution was used before the CPC at the Traffic Supersite during the measurements to get reliable results also during periods of high PNC at the site.

During two three-week periods between August 30$^{th}$ and October 10$^{th}$, 2022, a total of seven AQ Urban sensors were installed at the Traffic Supersite. One AQ Urban sensor was chosen as a reference instrument (Ref) since it was located at the Traffic Supersite during the whole six-week period. In addition to the reference instrument six other AQ Urban sensors were used (1−6). Three sensors during the first three weeks (1−3, August 30$^{th}$ – September 19$^{th}$) and another three




during the last three weeks (4−6, September 19th – October 10th). During these periods PNC was measured also by using
two CPCs with different cut-sizes. The CPCs were an Airmodus A20 CPC with a cut-size 5.4 nm and an Airmodus A20
CPC with a cut-size 10 nm. A dilution was used for both CPCs at the Traffic Supersite during the six-week comparison
period.
Particle number size distribution was measured at both stations with a Differential Mobility Particle sizer (DMPS) using
a Vienna type Differential Mobility Analyzers. At the Traffic Supersite an Airmodus A20 model CPC was used in the
DMPS system. At the UB Supersite the Twin DMPS had TSI model 3772 and 3756 CPCs. At the Traffic Supersite the
measured particle size range was between ~10 and ~800 nm and at the UB Supersite it was between 3 and ~800 nm.
The effect of regional and long-range transport of particulate matter to Helsinki metropolitan area can be seen by the
elevated $PM_{2.5}$ and BC concentrations measured at an urban rural site located in Luukki (60.3143 °N, 24.6846° E). Luukki
air quality measurement station is operated by the HSY and is a Helsinki metropolitan area rural background station
situated in clean background area 20 km from the Traffic Supersite. At the urban rural site, no major local pollution
sources are nearby and the increased concentrations of $PM_{2.5}$ and BC are mainly due to long range or regional transport
of particulate matter. The concentrations of $PM_{2.5}$ and BC at Luukki measurement station were measured using the Fidas
200 (Palas GmbH) and Multi-Angle Absorption Photometer (MAAP, Thermo Electron Corporation) instruments. The
further discussion is based on hourly-averaged data if not otherwise mentioned.
3   Results and Discussion
3.1   Comparison of particle number concentrations between AQ Urban sensors and CPCs
The boxplots of hourly-averaged PNC (particles $cm^{-3}$) measured with the different AQ sensors during the two instrument
comparison periods are shown in Fig. 1. The linear regression between the PNC measured with different AQ Urban
sensors (1-6) to the reference AQ Urban sensors are shown in Fig. S1. The Pearson correlation coefficient ($r$) is 0.99 for
all other AQ Urban sensors except for AQ Urban sensor 5 which had slightly lower correlation coefficient ($r$=0.97). This
is probably due to few outliers in the data set. The slope of the linear regression of the measured PNC between AQ Urban
sensors against the reference AQ Urban sensor varied between 1.0 and 1.06. The offset of the linear regression was
negative or positive depending on the AQ Urban sensor, but it was low compared to the measured PNC. The agreement
of PNC measured with different AQ Urban sensors can be concluded to be very good, which can be seen also from the
time series in Fig. S2.

Traffic Supersite

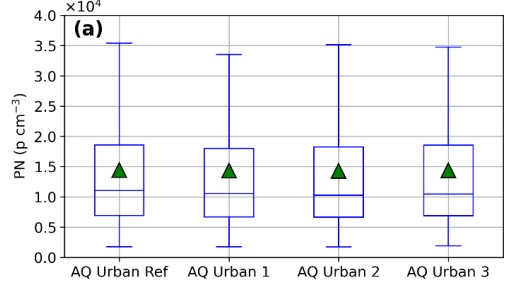
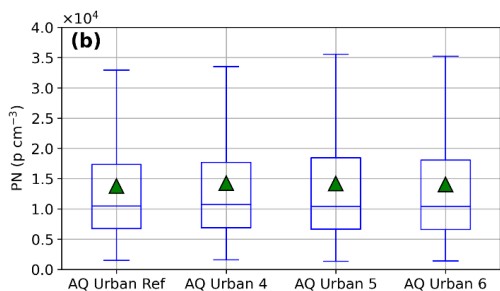




**Figure 1.** Comparison of hourly-averaged PNC measured with seven different AQ Urban sensors at the Traffic Supersite during the
first (a) and second (b) 3-week comparison period. The median is the horizontal line within the box, and the green triangle is the mean
value. The box spans from the first to the third quartile, and the whiskers extend to 1.5 times the interquartile range. The outliers are
not shown in the figure.
During the instrument comparison period PNC were measured at the Traffic Supersite also with two Airmodus A20 CPCs
having different cut-sizes, being 5.4 nm and 10 nm (Fig. 2). The measured hourly-averaged PNC with the reference AQ
Urban sensor agreed more closely to that measured with the CPC having a cut-off size of 10 nm compared to that having
a cut-off size of 5.4 nm. This is expected since 10 nm is the lower estimated detection limit of the AQ Urban sensor. The
correlation coefficients ($r$) between the measured PNC with the reference AQ Urban sensor and the CPCs having a cut
off diameter of 5.4 and 10 nm were 0.98 and 0.97 respectively (Fig. S3a and S3b). The slope of CPCs having cut-off
diameters 5.4 and 10 nm respect to AQ Urban sensors were 1.36 and 0.73 respectively (Fig. S3a and S3b). The lower
particle number concentrations measured with the AQ Urban compared to the CPC with the cut-off size 5.4 nm is due to
the different lower detection limits of these two instruments. Especially in the vicinity of heavily trafficked streets the
concentration of particles below 10 nm can expect to be high (e.g. Belkacem et al., 2020; Choi et al., 2014; Rönkkö and
Timonen, 2019) and has been measured to be significant also at the Traffic Supersite (Hietikko et al., 2018; Teinilä et al.,
2024). The lower slope of the linear regression between the CPC with cut-off size of 10 nm and to AQ Urban sensor
indicates that the AQ Urban sensor measures also particles slightly below 10 nm.

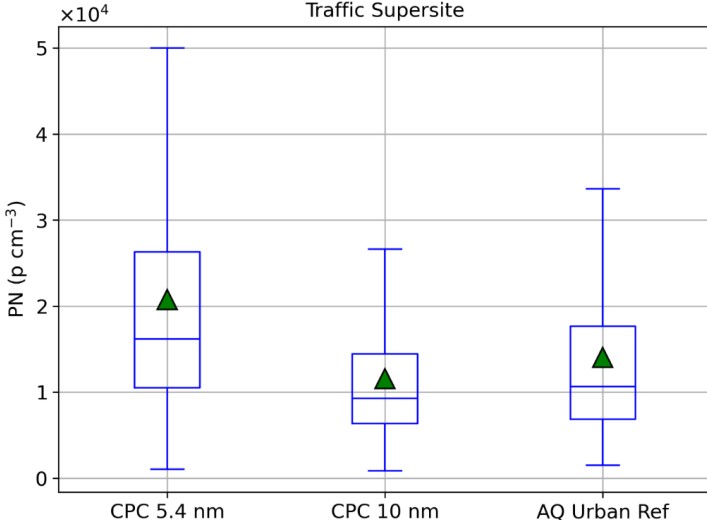


**Figure 2.** Comparison of hourly-averaged PNC measured with the reference AQ Urban sensor and two CPCs with different cut sizes
at the Traffic Supersite during the 6-week period. The median is the horizontal line within the box, and the green triangle is the mean
value. The box spans from the first to the third quartile, and the whiskers extend to 1.5 times the interquartile range. The outliers are
not shown in the figure.
The hourly diurnal variations of PNC measured with two CPCs with different cut sizes and the reference AQ Urban sensor
during the comparison period are shown in Fig. 3a and the differences in the measured PNC in Fig. 3b. The diurnal





patterns of the measured PNC with the different instruments are identical although the measured PNC are different
showing that they all observe the contribution of traffic on the PNC Especially, during the morning rush hours, the PNC
increased at the Traffic Supersite (Fig. 3a). The higher PNC during the morning rush hour was likely related to the more
efficient dilution and mixing of pollutants during afternoon. Also, the lanes of traffic towards the city center were closer
to the measurement station, which may emphasize the effects of morning rush hour when people are heading towards city
centre. The evidence of the existence of particles below 10 nm can be seen when comparing the two CPCs during the rush
hours. Also, the slope of the measured PNC between the CPC with cut-off size 5.4 nm with respect to the CPC with cut-
off size 10 nm was 1.80 (Fig. S3c). The AQ Urban sensor measured higher PNC compared to the CPC with the cut-off
size 10 nm during the rush hours. This result further supports the idea that the lower detection limit of AQ Urban sensor
was less than 10 nm. However, it is also possible that the AQ Urban sensor estimated the count median diameter (CMD)
erroneously (see the discussion in the next chapter).

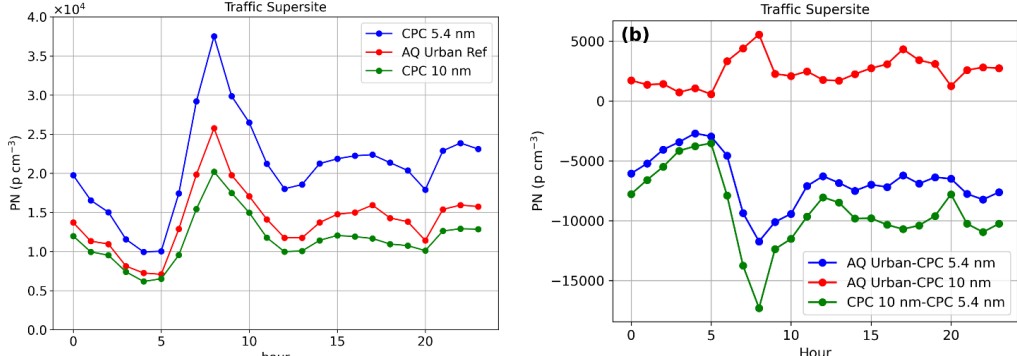


**Figure 3.** Hourly diurnal variation of the measured PNC (a) and the PNC difference (b) measured with two CPCs having different cut-
size and the reference AQ Urban sensor during the 6-week comparison period at the Traffic Supersite.

### 3.2    Particle number concentrations measured in two different environments in Helsinki

The PNC measured with the AQ Urban sensor and the CPCs in two different urban environments (Traffic Supersite and
UB Supersite) were compared between January 1$^{st}$ and August 15$^{th}$, 2022. At the UB Supersite the average PNC measured
with the AQ Urban sensor was like that measured with the CPC having a cut-of size 7 nm (Fig. 4). At the Traffic Supersite
the measured PNC with the AQ Urban sensor were lower compared to those measured with the CPC having a cut-off size
5.4 nm which was observed also during the six-week comparison period.



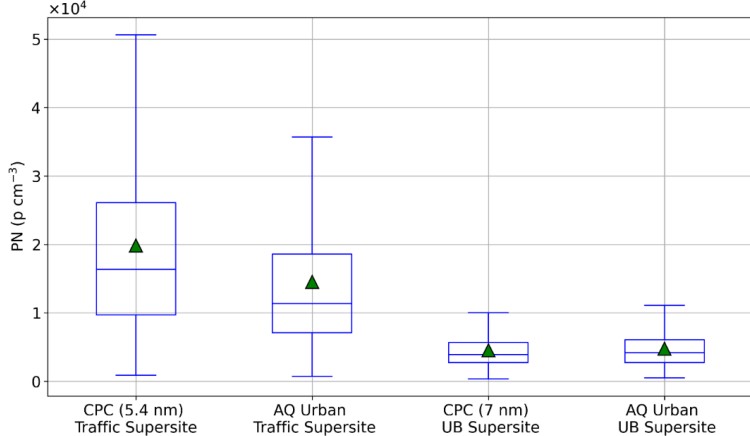


**Figure 4.** Comparison of hourly-averaged PNC measured with the AQ Urban and CPCs at the Traffic Supersite and at the UB Supersite during the 7.5-month measurement period. The cut-off size of the CPC at the Urban traffic site was 5.4 nm and at the Urban background site 7 nm. The median is the horizontal line within the box, and the green triangle is the mean value. The box spans from the first to the third quartile, and the whiskers extend to 1.5 times the interquartile range. The outliers are not shown in the figure.

The hourly diurnal variation of the PNC measured with the AQ Urban sensors and the CPCs at both sites are shown in Fig. 5a and the difference of the measured PNC between the AQ Urban sensors and the CPCs in Fig. 5b.

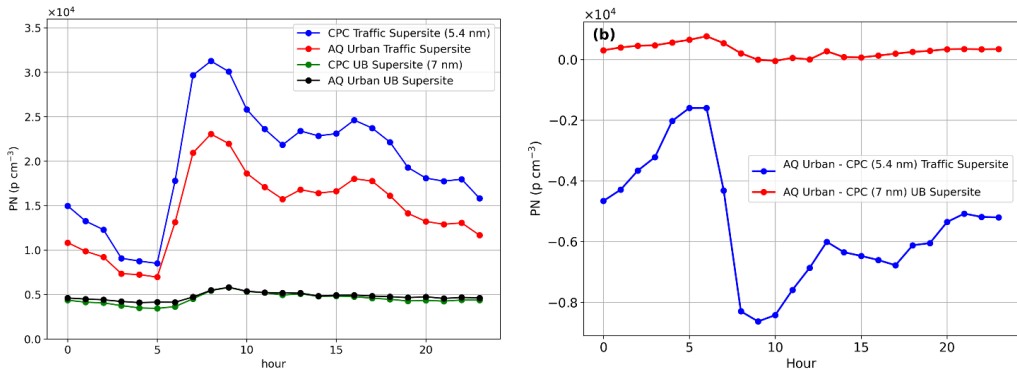

192

**Figure 5.** Hourly diurnal variation of measured PNC at the Traffic Supersite and at the UB Supersite with the AQ Urban sensors and the CPCs (a) and the difference of the measured PNC (b) during the 7.5-month measurement period. Notice the different cut-sizes of the CPCs.

At the UB Supersite the difference in the measured PNC with the AQ Urban sensor and CPC is close to zero throughout the day. At the Traffic Supersite this difference is negative throughout the day, and the difference starts to increase when the morning rush hour starts as was observed at the Traffic Supersite also during the comparison period. The difference in the cut-off size of the CPCs at these two sites was only 1.6 nm so it probably cannot explain the markedly higher differences of the two instruments. The observed difference is likely connected to the different concentrations and particle size distributions between these sites. This idea is supported by the hourly-averaged particle number size distributions in



Fig. S4 which show that at the Traffic Supersite the PNC for particles < 30 nm increase during the morning rush hour.
The DMPS data below 10 nm is not available from the Traffic Supersite, but the shape of the size distributions indicates
an increasing trend of PNC also below 10 nm size (Fig. S4a). On the other hand, the PNC shows decreasing trend during
all hours at the UB Supersite for particles < 10 nm (Fig. S4b). The size distribution results suggest that particles < 10 nm
do not considerably contribute at the UB Supersite. Hence, the much lower concentration of particles below 10 nm at the
UB Supersite is probably the main reason for the better agreement between AQ Urban and the CPC in terms of average
concentration.
The linear correlations of the measured PNC with the AQ Urban sensors and the CPCs at the Traffic Supersite and at the
UB Supersite are shown in Fig. 6, where the data set was divided based on the median of the Urban remote site $PM_{2.5}$
concentration  into two data sets; $PM_{2.5} < 2.5\ \mu g\ m^{-3}$ and $PM_{2.5} > 2.5\ \mu g\ m^{-3}$. The slope of the linear regression was
slightly higher (1.24, $r = 0.94$) compared to the whole data set (1.17, $r = 0.93$, Fig. S5) at the Traffic Supersite when $PM_{2.5}$
concentration at the Urban remote site was elevated and lower (1.09, $r = 0.93$) when its concentration was low. At the UB
Supersite the slope of the linear regression was same for the whole data set (0.83, $r = 0.89$, Fig. S5) and for the data set
where the low Urban remote $PM_{2.5}$ points were discarded ($r = 0.86$). However, when discarding the data points with high
$PM_{2.5}$ concentration at the Urban remote site, the slope increased from 0.83 to 0.93 together with increasing correlation
coefficient ($r$) which increased from 0.86 to 0.96. These results suggest that the PNC measurement of the AQ Urban was
affected by the regional background $PM_{2.5}$ concentrations. This idea is supported also by the colored scatter plots in Fig.
6 (and Fig. S5a and S5c), where the slope of this linear correlation seemed to be dependent on the $PM_{2.5}$ concentration
measured at the Urban remote site. The daily-averaged time series of $PM_{2.5}$ in Fig. S6 shows that the periods of elevated
$PM_{2.5}$ concentrations are typically seen at all sites, indicating either regional or long-range transportation. During the
winter BC concentration at the UB remote site increases simultaneously with the $PM_{2.5}$ concentration.
In Figs. S5b and 5d, the scatter plots, colored by the measured concentrations of $NO_x$ at the same sites where the PNC
were measured, are shown. At the Traffic Supersite higher PNC were measured together with high $NO_x$ concentrations
due to their common source (motor vehicle emissions, Fig. S5). However, the linear correlation of PNC measured with
the AQ Urban sensor and CPC was similar despite the varying $NO_x$ concentrations at least at the Traffic Supersite (Fig.
S5a).




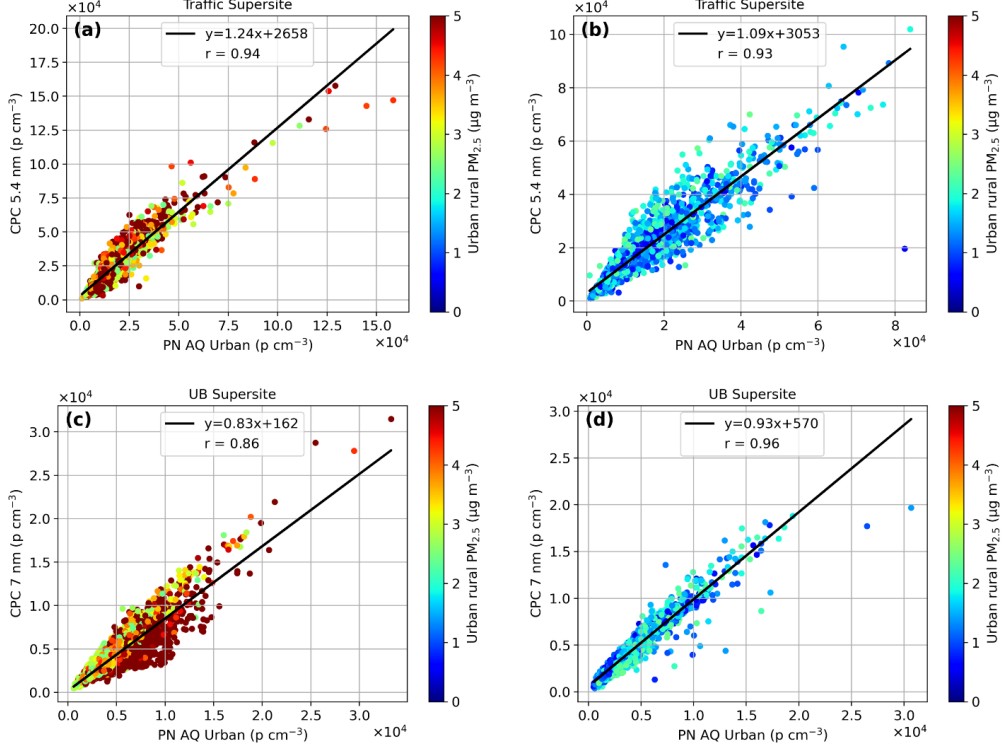


**Figure 6.** Linear correlation of hourly-averaged particle number concentrations measured with the AQ Urban sensor and CPC at the Traffic Supersite and at the UB supersite during high (a and c) and low (b and d) $PM_{2.5}$ concentrations at the Urban remote site during the 7.5-month measurement period. The color of the markers indicates $PM_{2.5}$ concentration at the Urban remote site.

The results in Fig 6 and Fig. S5 show the agreement with the measured PNC between AQ Urban sensors and CPCs seem to be near unity when discarding the periods with regional or long-range transport (high $PM_{2.5}$ at the Urban remote site). The transported particles are aged and have larger sizes, and they mix externally with the traffic related ultrafine particles forming bimodal particle size distribution, potentially affecting the performance of the AQ Urban. Interestingly, however, the effect of increased regional $PM_{2.5}$ seems to be different at the Traffic Supersite (increased $PM_{2.5}$ increased the slope) compared to the UB Supersite (increased $PM_{2.5}$ decreased the slope).

The monthly-averaged particle size distributions in Figs. 7a and 7b show that the size distributions shifted towards larger particle size during the measurement period and the bimodal structure of the particle size distribution became clearer. The mean particle size increased towards the summer months, which can be seen also when looking the daily-averaged time series in Fig. S7, where the count median diameter from the AQ Urban sensor and the mean particle diameter calculated from the DMPS data increased toward the summer months. Like the effect of $PM_{2.5}$ in Fig 6 and Fig S5, the increased bimodality and mean particle diameter affected the ratio of the PNC measured with the CPCs and the AQ Urban sensors (Fig. S7). However, the ratio between CPCs and AQ Urbans increased at the Traffic Supersite and decreased at the UB



Supersite. In general, it is not clear why the particle size increases during summer months. The possible reason may be
the growth of particles due to more favorable secondary aerosol formation via oxidation of organic matter from motor
vehicle exhaust during summer months (e.g. Ahlm et al., 2012; Gentner et al., 2017, 2012). During the summer months
the increased solar radiation, increased water content (Fig. S8) and increased concentrations of biogenic organic matter
may be another explanation for this growth (e.g. Srivastava et al., 2022).

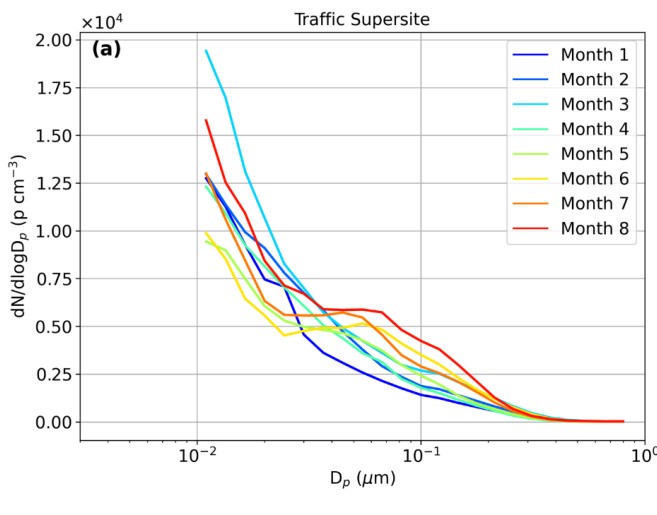

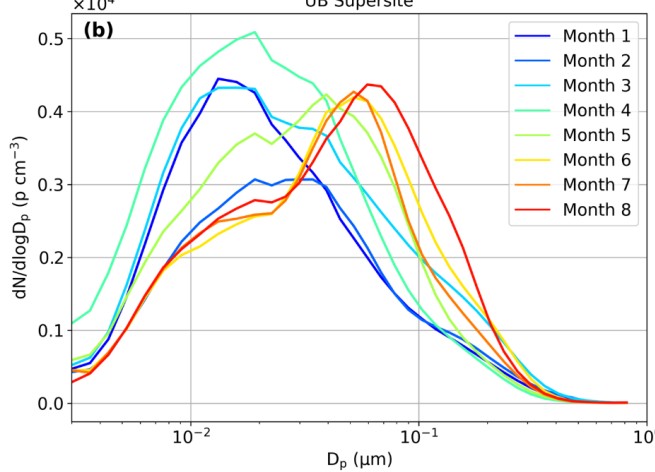


**Figure 7.** Monthly-averaged particle number size distributions measured with DMPS at the Traffic Supersite (a) and at the UB Supersite
(b) during the 7.5-month measurement period.
Results on the effects of regional $PM_{2.5}$ (Fig. 6 and Fig. S5) and the seasonality (Figs. 7 and S7) both suggest that the
effect of bimodal particle size distribution on the performance of AQ Urban is different. This finding could be explained
by the varying particle characteristics at these microenvironments. At the Traffic Supersite the PNC was constantly high



due to the road traffic. During the regional and long-range transport periods the observed particle size distribution was
not anymore unimodal due to the external mixing of traffic-related and regional and long-range transported particles. The
AQ Urban sensor estimates the count median diameter using the assumption that the particle size distribution is unimodal.
Therefore, the increased regional background concentration causes an increase in the estimated count median diameter,
which reduces the conversion factor used to convert the electric current to PNC. The increased accumulation mode,
however, does not considerably affect the total PNC, which is still dominated by particles smaller than 20 nm (Fig. 7a).
Hence, even though the accumulation mode particles also affect the electric current measured by the AQ Urban, the
decreased conversion factor due to the increased estimated CMD, causes the sensor to underestimate the PNC at the
highly trafficked site.
At the UB Supersite the effect of traffic was low compared to traffic supersite, and the particle size distribution was
constantly concentrated on larger particle sizes (Fig. 7b). When particle size distribution was shifted to even larger particle
sizes (during summer or pollution transportation event) the AQ Urban sensor measures higher electric current. The
maximum limit of the estimated count median diameter of the AQ Urban sensor is ~100 nm but during the regional or
long-range transport periods, the size of the particles contributing the most to the measured electric current may be above
this limit, and so the AQ Urban sensor overestimates PNC.
Overall, the effect of bimodal size distribution on the AQ Urban measurement is important to consider when conducting
measurements in varying urban environments and geographic regions. As seen in the results, the performance of AQ
Urban in the PNC measurement was mainly very good in our measurements. However, the regional particle concentration,
thus, the accumulation mode of particles, was typically very low, which seemed to be especially suitable for the
performance of AQ Urban. It should be noted that the uncertainty caused by the bimodal size distribution could be much
more significant in locations where regional background concentrations are higher, like Central/Eastern Europe or India
(Sebastian et al., 2022; Trechera et al., 2023). Also, some particle sources, like residential wood combustion, can
considerably contribute to concentrations of particles larger than 100 nm (Harni et al., 2023; Kalkavouras et al., 2024)
potentially causing similar challenges as the increased regional background concentration. In general, it's, however, worth
noting that the measurement principle of AQ Urban is rather like other diffusion charger-based PNC sensors, like the
Partector 2 (Asbach et al., 2024). The challenge of bimodal size distribution has also been observed earlier when
considering the LDSA measurement of the diffusion charger-based sensors (Lepistö et al., 2024). Hence, it is justifiable
to think that the challenge related to bimodal size distribution could be relevant for other diffusion-charger based PNC
sensors as well.
Data Availability
Data available on request.
**4    Conclusions**
We investigated the possibility of using AQ Urban sensors in urban air quality monitoring to obtain PNC. The comparisons
were made at two different sites, at a heavily trafficked street canyon (Traffic Supersite) and at an urban background site
(UB Supersite) in 2022. First, the agreement between different AQ Urban units were investigated in two three-week
lasting campaigns (August 30th and October 10th, 2022) in the Traffic Supersite: the agreement with the measured particle



number concentrations within total of seven different AQ Urban sensors was good (Pearson *r*: 0.97–0.99, linear fit slopes:
1.0–1.06), showing that results with different AQ Urban units are well comparable in general. During this comparison
period, the PNC measured with a reference AQ Urban sensor were also compared to those measured with two CPCs
having cut-off sizes 5.4 and 10 nm. On average, the PNC measured with the AQ Urban sensor were slightly higher than
those measured with the CPC having a cut-off size 10 nm. The relative difference was, however, low compared to the
measured PNC. Also, the correlation between the reference AQ Urban sensor and the 5.4 nm and 10 nm CPCs were 0.98
and 0.97, respectively. These findings show that AQ Urban sensors should be well-suitable to measure the concentration
of particles approx. larger than 10 nm in highly trafficked areas.
The long-term agreement between AQ Urban sensors and CPCs was also investigated at the Traffic Supersite and UB
Supersite (January 1st and August 15th, 2022). Overall, the correlation between AQ Urban sensors and the CPCs was good
at both sites (*r* being 0.93 and 0.89, respectively), even though the cut-off sizes of the CPC at these two sites were different
(5.4 and 7 nm) compared to the lower limit of AQ Urban sensors (approx. 10 nm). The difference between AQ Urban
sensors and the CPCs increased especially during traffic rush hours at the Traffic Supersite. This result can, however, be
explained because of the increased emissions of particles smaller than 10 nm from traffic which are not detected with the
AQ Urban sensor. On the other hand, it was noted that especially long-range transported (LRT) pollution episodes as well
as time of the year can affect the accuracy of AQ Urban sensors. This result can be explained by the bi-modal particle size
number distributions observed especially during the LRT-episodes and summer, because AQ Urban sensor estimates the
count median diameter of particles assuming that the particle number size distribution is unimodal. Hence, the conversion
from the detected electric current into PNC cannot accurately estimate the size of the detected particles, causing
uncertainty in the measurement. Despite this downside of the method, it should be noted that the correlation between the
AQ Urban sensor and the CPCs was good during the whole measurement period, indicating that the sensor is well-suitable
for long-term particle number concentration monitoring in Helsinki.
Overall, the results show that AQ Urban sensor was well suitable to measure the number concentration of particles
approximately larger than 10 nm in two different urban environments in Helsinki, Finland. The result is interesting
regarding the EU's new air quality directive (2024) which requires particle number (> 10 nm) concentration monitoring
at pollution hotspots. The results show that diffusion charger-based measurement of PNC should be well suitable for
urban air quality monitoring, enabling more-dense sensor monitoring network than CPC methodology. For example, the
sensors could be utilized to estimate potential hotspots for the measurements required by the directive. Still, it should be
noted that further validation of diffusion charger-based particle number measurements is needed. Even though the
challenges caused by bi-modal particle number size distributions in this study were rather minimal, it needs to be
considered that Finland has very clean air in terms of regional pollution (e.g., $PM_{2.5}$). Hence, the challenges caused by bi-
modal particle number size distribution could be much more significant in locations with higher regional pollution, i.e.,
higher accumulation mode of particles. Thus, further studies of the performance of diffusion charger-based particle
number sensors from different locations would be valuable for further conclusions.
Competing interests
The authors declare that they have no competing interests.



Acknowledgements
This work was supported by the Technology Industries of Finland Centennial Foundation to Urban Air Quality 2.0 project,
the EU Horizon 2020 Framework Programme via the Research Infrastructures Services Reinforcing Air Quality
Monitoring Capacities in European Urban & Industrial Areas (RI-URBANS) project (GA-101036245), European Union's
Horizon Europe research and innovation programme under grant agreement No 101096133 (PAREMPI: Particle emission
prevention and impact: from real-world emissions of traffic to secondary PM of urban air), GIANT (Global trends in IAQ:
Novel technologies, Competence and Business, nr 4736/31/2023)) funded by Business Finland and participating
companies, and the Academy of Finland ACCC Flagship (grant no. 337552, 337551, 337549).

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
