# Peer review of "Comparison of particle number concentrations measured with AQ Urban sensors in two different environments in Helsinki, Finland"

_EGUsphere, 2025_

## Referee Comment (RC1)

This paper presents interesting data and valuable findings using alternative technologies to measure both particle number and size distribution. The overall structure is coherent, the narrative is clear, and the study is highly relevant to current discussions on particle number monitoring and instrumentation. The authors provide a balanced evaluation of the benefits and limitations of the data and methods investigated, and the work is generally well-presented and discussed.

However, I am recommending Major Revisions. This is primarily due to issues with the Supplementary Material: several referenced figures (notably S3c and S4) are missing or incorrectly referenced, which makes it difficult to follow key arguments related to particle size distributions and sensor cut-off behavior. These figures are essential for following or backing up the claims presented in the paper, and without them, the supporting evidence cannot be properly evaluated.

Additionally, the treatment of "outliers" requires a more rigorous and transparent explanation. The current description does not sufficiently clarify their origin, impact, or the rationale behind their exclusion.

Specific revision points are listed below.

Line 53-54: the sentence begins by explicitly stating "According to EU directive 2024..." but then ends with the phrase "sites with high" and the formal citation "(EU) 2024/2881 (2024)" , this is difficult to read and is left with a possible missing end to the paragraph, a clearer approach would be to integrate the citation by writing "According to Directive (EU) 2024/2881, PNC monitoring is regulated at..." and include the missing text "sites with high ultrafine particle (UFP) concentrations"

Line 60-62: could do with being smother and a bit mixed and repetitive. I recommend rephrasing for clarity and flow, with perhaps. "CPC measurements are costly (due to instrument purchase and servicing), and maintaining the system is time-consuming, requiring regular maintenance and frequent butanol refills."

Line 67: incorrect comma usage in "instruments, (AQ™ Urban sensors..." so should be "instruments (AQ™ Urban sensors..."

Line 104-107: 1) not clear if there were there any side-by-side comparisons of the Airmodus A20 and the TSI 3756 to quantify any instrumental bias between the two reference devices before deployment and if so was this applied to the final data? 2) at the traffic supersite how was the dilution applied to the instrument and how was the dilution factor determined and was it stable throughout the measurement campaign? 3) what quality assurance or control procedures were applied to the instruments during the campaign.

Line 132: mentions "Fig. S1" is the "S" to refer to an additional supplementary figure not supplied/drafting format or a miss type as line 140 refers to Figure 1 which I assume is the intention so perhaps change to "Fig. 1"

Line 134: The text states that "AQ Urban sensor 5" showed a lower correlation due to "a few outliers," and uses the word probably, suggesting the cause has not been clearly identified. Since Figure 1 explicitly excludes outliers from the visualization, its important to clarify the nature of these data points. Were they caused by instrumental issues (e.g., electrometer noise, power interruptions, calibration drift) or by genuine transient high-concentration events? Distinguishing between sensor malfunction and true environmental variability is important for assessing operational reliability and interpreting the final conclusions.

Line 138: "Fig. S2" to "Fig. 2" if comment in Line 132 is the same?

Line 144: missing comma between the text "period PNC"

Line 145: remove the comma after "cut-sizes" and rephrase with "cut-sizes of 5.4 nm and 10 nm (Fig. 2)."

Line 149 and 150: "Fig. S3a and S3b" to "Fig. 3a and 3b" if comment in Line 132 is the same?

Line 155-156: The statement that the 'AQ Urban sensor measures also particles slightly below 10 nm' should be rephrased. As noted in the Experimental section (Lines 96–101), the sensor measures electrical current and calculates PNC based on an assumed unimodal lognormal distribution. It would be more precise to state that the sensor 'detects the charge fraction of particles below 10 nm' or that the 'PNC calculation accounts for the influence of particles <10 nm,' rather than implying a direct counting measurement of them.

Line 171: unable to locate "Fig S3c" in the article, is this a supplementary figure missing or is it missing from figure 3? As mentioned in comment on line 132.

Line 182: "cut-of size" please correct to "cut-off size"

Line 201–208 (Missing Supplementary Data): The text references "Fig. S4", "Fig. S4a", and "Fig. S4b" to support claims regarding hourly-averaged particle number size distributions. Specifically, the text states that Figure S4 shows an increase in particles <30 nm at the Traffic Supersite during morning rush hours and a decreasing trend for particles <10 nm at the UB Supersite.

However, this figure appears to be missing.

Figure 5 shows only the diurnal variation of the total PNC, not the size distribution.

Figure 6 shows hourly average particle number concentrations

Figure 7 shows the monthly-averaged size distributions, which does not provide the hourly resolution required to verify the rush-hour claims.

Since this section relies on this specific DMPS data to explain the discrepancy between the AQ Urban sensor and the reference CPCs (attributing it to the <10 nm fraction), it is critical that these hourly size distribution plots are provided. Please verify the Supplementary Material citations and ensure Figures S4a and S4b are included or correct the figures accordingly id required.

Line 120, the station is referred to as "urban rural site" but in line 210 its referred to as "Urban remote site" correct the phrasing to maintain consistency

Line 212, 214, 219, 223, 225, 227, 233, 243, 254: referred to supplementary "Fig. S5", "Fig. S5a" and "Fig. S5b" please review text description or supply the figures if not referring to Figure 5.

Line 220: "Fig S6" please review text description or supply the figures if not referring to Figure 6.

Line 245, 254: "Fig S7" please review text description or supply the figures if not referring to Figure 7.

Line 249: text refers to "Fig. S8" which is missing

Line 251-252: Regarding Figure 7: The x-axis currently displays particle diameter (Dp) in micrometers (µm) using a log scale (e.g., $10^{-2}$). However, the discussion in the text (Lines 254–265) and throughout the manuscript refers to particle sizes exclusively in nanometers (e.g., 'particles smaller than 20 nm' in Line 261, '~100 nm' in Line 268). To improve readability and allow for easier cross-referencing between the text and the visual data, I recommend changing the x-axis units in Figure 7 to nanometers (nm).

Line 342: Citing an EU Directive as "Anon" (Anonymous) is non-standard for academic journals. It is usually better to cite the author as "European Union", "European Parliament", or "European Commission"

Line 362: The citation for the CEN standard is listed as 'EN 16976:3035'. This appears to be a typo. Based on Line 56, this should refer to 'EN 16976:2024'. Additionally, please correct the spelling of 'Standartization' to 'Standardization' on line 363.

Line 385: doesn't appear to be a complete citation

Line 389-381: citation missing the journal name: "Boreal Environmental Research"

Line 414: citation missing the journal name: "Aerosol Research, 2, 271-289"